# Effects of Alirocumab on Triglyceride Metabolism: A Fat-Tolerance Test and Nuclear Magnetic Resonance Spectroscopy Study

**DOI:** 10.3390/biomedicines10010193

**Published:** 2022-01-17

**Authors:** Thomas Metzner, Deborah R. Leitner, Karin Mellitzer, Andrea Beck, Harald Sourij, Tatjana Stojakovic, Gernot Reishofer, Winfried März, Ulf Landmesser, Hubert Scharnagl, Hermann Toplak, Günther Silbernagel

**Affiliations:** 1Department of Internal Medicine, Division of Angiology, Medical University of Graz, 8036 Graz, Austria; metzner.thomas@gmx.at (T.M.); guenther.silbernagel@medunigraz.at (G.S.); 2Department of Medical Affairs, Sanofi-Aventis GmbH, 1100 Vienna, Austria; 3Department of Internal Medicine, Division of Endocrinology and Diabetology, Medical University of Graz, 8036 Graz, Austria; deborah.leitner@gmx.at (D.R.L.); karin.mellitzer@yahoo.de (K.M.); Andrea.Beck@klinikum-graz.at (A.B.); ha.sourij@medunigraz.at (H.S.); 4Clinical Institute of Medical and Chemical Laboratory Diagnostics, University Hospital Graz, 8036 Graz, Austria; stojakovic@gmx.at; 5Department of Radiology, Clinical Division of Neuroradiology, Vascular and Interventional Radiology, Medical University of Graz, 8036 Graz, Austria; gernot.reishofer@medunigraz.at; 6Department of Internal Medicine 5 (Nephrology, Hypertensiology, Endocrinology, Diabetology, Rheumatology), Mannheim Medical Faculty, University of Heidelberg, 68167 Mannheim, Germany; winfried.maerz@synlab.com; 7Synlab Academy, Synlab Holding Germany GmbH, 86156 Augsburg, Germany; 8Clinical Institute of Medical and Chemical Laboratory Diagnostics, Medical University of Graz, 8036 Graz, Austria; hubert.scharnagl@medunigraz.at; 9German Center for Cardiovascular Research (DZHK)-Partner Site Berlin, Department of Cardiology, Berlin Institute of Health, Charité University Medicine Berlin, 12200 Berlin, Germany; ulf.landmesser@charite.de

**Keywords:** alirocumab, PCSK9, lipids, nuclear magnetic resonance spectroscopy, fat-tolerance test

## Abstract

Background: PCSK9 antibodies strongly reduce LDL cholesterol. The effects of PCSK9 antibodies on triglyceride metabolism are less pronounced. The present study aimed to investigate in detail the effects of alirocumab on triglycerides, triglyceride-rich lipoproteins, and lipase regulators. Methods: A total of 24 patients with an indication for treatment with PCSK9 antibodies were recruited. There were two visits at the study site: the first before initiation of treatment with alirocumab and the second after 10 weeks of treatment. Fat-tolerance tests, nuclear magnetic resonance spectroscopy, and enzyme-linked immunosorbent assays were performed to analyze lipid metabolism. Results: A total of 21 participants underwent the first and second investigation. Among these, two participants only received alirocumab twice and 19 patients completed the trial per protocol. All of them had atherosclerotic vascular disease. There was no significant effect of alirocumab treatment on fasting triglycerides, post-prandial triglycerides, or lipoprotein-lipase regulating proteins. Total, large, and small LDL particle concentrations decreased, while the HDL particle concentration increased (all *p* < 0.001). Mean total circulating PCSK9 markedly increased in response to alirocumab treatment (*p* < 0.001). Whereas PCSK9 increased more than three-fold in all 19 compliant patients, it remained unchanged in those two patients with two injections only. Conclusion: Significant effects of alirocumab on triglyceride metabolism were not detectable in the ALIROCKS trial. The total circulating PCSK9 concentration might be a useful biomarker to differentiate non-adherence from non-response to PCSK9 antibodies.

## 1. Introduction

Low-density lipoprotein (LDL) cholesterol causes atherosclerotic cardiovascular disease [1]. The magnitude of life-long LDL cholesterol exposure is a major determinant of individual cardiovascular risk [2]. The earlier and more strongly the LDL cholesterol level is reduced, the greater the reduction in cardiovascular risk [3]. However, despite substantial reductions in LDL cholesterol with statins and newer lipid-lowering agents, patients continue to experience recurrent major cardiovascular events [4].

Accumulating evidence supports the relevance of triglyceride-rich lipoproteins in the management of cardiovascular disease [5,6]. Elevations of circulating triglyceride-rich lipoproteins are the result of their overproduction in the liver and/or their reduced clearance from the circulation [7]. As apolipoprotein B (ApoB)-containing lipoproteins, they carry considerable amounts of cholesterol and can enter the subintimal space, especially as remnants after the hydrolysis of triglycerides by lipoprotein lipase [5,6]. The subintimal deposition of cholesterol then leads to atherosclerotic plaques [1]. Fibrates, which reduce triglyceride-rich lipoproteins via activation of peroxisome proliferator activated receptor α, have been effective in reducing cardiovascular risk in patients with high triglycerides [8]. Accordingly, the REDUCE-IT trial showed that treatment with icosapent-ethyl, an omega-3 fatty acid, significantly reduced triglycerides and cardiovascular events [9]. Further clinical trials evaluating the safety and efficacy of medical therapies that target key regulators of triglyceride-rich lipoprotein metabolism are currently ongoing [10,11,12]. These key regulators include apolipoprotein C-III (ApoCIII), a potent inhibitor of lipoprotein lipase, and its antagonist apolipoprotein C-II (ApoCII) [11,13]. Angiopoietin-like proteins 3 and 4 (ANGPTL-3/4) inhibit lipoprotein lipase by forming complexes with angiopoietin-like protein 8 (ANGPTL-8), while glycosylphosphatidylinositol-anchored high-density lipoprotein-binding protein-1 (GPIHBP-1) facilitates the transfer of lipoprotein lipase to the luminal side of the endothelium, thereby increasing its catalytic activity to hydrolyze plasma triglycerides [10,12].

In addition to triglyceride-rich lipoproteins, high concentrations of lipoprotein a (Lp(a)) make up a considerable proportion of the residual cardiovascular risk. This statement is based on very solid genetic evidence from Mendelian randomization studies as the plasma concentration of this small lipoprotein is highly heritable [14,15,16].

Moreover, the lipoprotein particle distribution has been associated with cardiovascular risk [1,17]. In this respect, a high concentration of small dense LDL particles abundant in diabetic dyslipidemia is considered atherogenic as they can easily penetrate the endothelium to be deposited in the subintimal space [18,19,20]. On the other hand, high concentrations of small high-density lipoprotein (HDL) particles are considered especially protective as they are considered effective vehicles for reverse cholesterol transport [21,22].

The effects of novel proprotein convertase subtilisin/kexin type 9 (PCSK9) antibodies on LDL cholesterol and cardiovascular event reduction have thoroughly been investigated [23,24]. However, less evidence is available on the effects of PCSK9 antibodies on triglyceride metabolism and lipoprotein particle concentrations.

Therefore, the present study aimed to evaluate in detail the effects of the fully human PCSK9 antibody alirocumab on triglyceride-rich lipoproteins and lipoprotein particle subfractions. Moreover, the objective was to test the effects of treatment with alirocumab on lipoprotein lipase regulators. To answer these questions, we performed standardized fat-tolerance tests, nuclear magnetic resonance spectroscopy, lipoprotein analysis using a combined ultracentrifugation precipitation method, and enzyme-linked immunosorbent assays.

## 2. Methods

### 2.1. Study Design

The ALIROCKS study (NCT03559309) is a monocentric, longitudinal pilot study with 24 patients scheduled for treatment with alirocumab. Treatment with alirocumab was initiated based on the current guidelines for the treatment of dyslipidemia of the European Atherosclerosis Society and the European Society of Cardiology [25]. Inclusion criteria were atherosclerotic cardiovascular disease and elevated LDL cholesterol despite maximum tolerated oral lipid-lowering therapy. Exclusion criteria were age < 18 years, pregnancy, and previous treatment with PCSK9 antibodies. There were 2 visits at the study site, the first before initiation of treatment with alirocumab and the second after 10 weeks of treatment. Diagnostic procedures of the first visit included a physical examination and a questionnaire on the medical history. Laboratory and fat-tolerance testing were performed at the first and second visit. Predefined secondary outcome measures included changes in lipoprotein particle distribution, composition, and post-prandial lipemia, in response to treatment with alirocumab. The clinical trial was conducted in accordance with the Declaration of Helsinki and was approved by the Austrian Federal Office for Safety in Health Care (EudraCT: 2018-000981-12) and the Ethics Committee of the Medical University of Graz (protocol number: 29-519 ex 16/17, approval date: 6 April 2018). All participants gave written informed consent.

### 2.2. Clinical Characterization

Information on demographics, clinical characteristics, and lipid medication was collected at inclusion in the study. Familial hypercholesterolemia was documented according to medical history. Patients not receiving high-intensity statins defined as ≥40 mg atorvastatin or ≥20 mg rosuvastatin at baseline were categorized as having partial or complete statin intolerance.

### 2.3. Fat-Tolerance Test

The Lipotest^®^ standardized meal was used for the fat-tolerance tests (D. GENOMERES Advanced Medical Research, Athens, Greece). It is a diagnostic meal with a special medical purpose, characterized as “food for specialized diagnostic determination of postprandial triglycerides”, and consists of 832 kcal total energy, comprising 75 g saturated fat, 25 g carbohydrates, 10 g protein, 2.1 g fiber, and 0.15 g salt [26]. The Lipotest^®^ meal was provided to the study center in a sachet and comprised 115 g powder that was rehydrated by adding 150 mL water. The water was poured in an adequate bowl first; afterwards, the powder of the sachet was added to the water and mixed with a hand mixer or a whisk until the powder was completely dissolved and became a homogenous drink. The patients were advised to fast for at least 10 h before the test. Laboratory testing was performed before the test meal and two and four hours after the test meal. LDL cholesterol during the fat-tolerance tests was measured by lipoprotein electrophoresis at T = 0 (0 h) and calculated according to the Friedewald equation at 2 and 4 h [27].

### 2.4. Lipoprotein Analysis by Ultracentrifugation and Precipitation

Lipoprotein particles were separated using a combined ultracentrifugation precipitation method [28,29,30]. Plasma was ultracentrifuged at a density of d = 1.0063 kg/L, at 30.000 rpm for 18 h. The very low-density lipoprotein (VLDL) particle fraction was removed, and LDL were precipitated with phosphotungstic acid/MgCl_2_. Cholesterol and triglycerides were measured using enzymatic reagents from Diasys (Holzheim, Germany) and were calibrated using secondary standards from Roche Diagnostics (Mannheim, Germany). Apolipoproteins (Apo), ApoB in VLDL and LDL, and Lp(a) were determined by immunoturbidimetry using reagents from DiaSys (Holzheim, Germany) and standards from Siemens (Marburg, Germany; ApoAI, ApoB, ApoE), Kamiya Biomedical (Seattle, WA, USA; ApoAII, ApoCII, ApoCIII), and Diasys (Lp(a)). The coefficients of variation were <5% [31].

### 2.5. Enzyme-Linked Immunosorbent Assays

ANGPTL-3 (ID: JP27750) and ANGPTL-4 (ID: JP27749) were measured with reagents from IBL (Minneapolis, MN, USA), GPIHBP-1 from TECAN (Männedorf, Switzerland, ID: 30131804), and circulating PCSK9 from Bio-Techne (Minneapolis, MN, USA, ID: DPC900). The PCSK9 assay measured total plasma PCSK9 irrespective of binding to the antibody alirocumab.

### 2.6. Nuclear Magnetic Resonance Spectroscopy

Analyses of lipoprotein particle subclasses were performed using the AXINON^®^ lipoFIT^®^ nuclear magnetic resonance assay (Numares, Regensburg, Germany) with an Avance III HD nuclear magnetic resonance spectrometer (Bruker; Billerica, MA, USA), an Ascend 600 MHz magnet (Bruker), and using TopSpin 3.2 (Bruker) and Axinon Suite 1.0.0.1 (Numares, Regensburg, Germany) software. The lipoprotein particle sizes were 50-240 nm for large VLDL particles (L-VLDLp), 21.2-23 nm for large LDL particles (L-LDLp), 18-21.2 nm for small LDL particles (S-LDLp), 8.8-13 nm for large HDL particles (L-HDLp), and 7.3-8.8 nm for small HDL particles (S-HDLp). Total and small VLDL particle concentrations are not provided by the Numares method (AXINON^®^ lipoFIT^®^ nuclear magnetic resonance assay, Appendix A) [32].

### 2.7. Statistical Analysis

The statistical design of this pilot trial was explorative and descriptive. The baseline characteristics are specified as numbers (percentages) in cases of categorical variables and as means (standard deviations) in cases of continuous variables. The area under the curve (AUC) analysis concerning the fat-tolerance testing was calculated by the trapezoid approximation (AUC^(T0−4)^ = AUC^(T0−2)^ + AUC^(T2−4)^; AUC^(T0−2)^ = (Value^(T0)^ + Value^(T2)^)/2 × (Time^(T2)^-Time^(T0)^), AUC^(T2−4)^ = (Value^(T2)^ + Value^(T4)^)/2 × (Time^(T4)^-Time^(T2)^). Changes between baseline and week 10 were tested with the paired-samples *t*-test and/or related samples Wilcoxon signed rank test when Shapiro–Wilk analysis suggested non-parametric distribution. *p*-values ≤ 0.05 were considered significant. The SPSS 26.0 statistical program was used (SPSS Inc., Chicago, IL, USA). 

## 3. Results

### 3.1. Study Population

The recruitment of the 24 participants was performed from 20 June 2018 to 30 January 2020 (Appendix A). Approximately 80% of the participants received 75 mg of alirocumab every two weeks and 20% received 150 mg alirocumab every two weeks. Two patients only received alirocumab injections twice, at baseline and at week 2. The reasons for their non-adherence were documented as drug access and communication barriers. Four patients had minor changes in their oral lipid-lowering therapies during the trial. Among these, two patients stopped their red yeast rice product, one stopped ezetimibe, and another one reduced their oral combination therapy from 80 mg atorvastatin plus 10 mg ezetimibe to 20 mg atorvastatin monotherapy. Due to the COVID-19 pandemic and the subsequent implementation of national countermeasures restricting all hospital visits to essential interventions only, the clinical trial had to be terminated before the last three participants had completed the study. Hence, 19 patients completed the trial (Table 1). The mean (standard deviation) treatment duration with alirocumab for participants who had completed the trial was 10.0 (0.7) weeks.

### 3.2. Characteristics of Trial Participants

The cohort included middle-aged and elderly females and males. All 19 participants had documented coronary artery disease. Among these, seven patients had poly-vascular disease with an additional diagnosis of cerebral and/or peripheral artery disease. The vast majority, 16 participants, had received coronary intervention or bypass surgery. All three patients without prior coronary intervention or surgery had poly-vascular disease with coronary stenosis (50%) confirmed by cardiac computed tomography and carotid artery occlusive disease. One of the three patients without prior coronary intervention or surgery also had peripheral artery disease. Familial hypercholesterolemia was documented in two patients, and type-2 diabetes mellitus in four patients. A total of 16 study participants had partial or complete statin intolerance. Hence, very few participants received high-intensity statins at baseline. However, the proportion of patients taking ezetimibe as concomitant medication was high (Table 1 and Appendix A).

There was a significant positive association of baseline small dense LDL particles and triglyceride-rich large VLDL particles (r = 0.883, *p* < 0.001). Small and large LDL particles were not significantly correlated (r = 0.021, *p* = 0.932).

### 3.3. Effects of Alirocumab on Post-Prandial Lipemia

Fourteen subjects participated in the fat-tolerance testing. Triglycerides were significantly increased by the Lipotest^®^ meal before and after 10 weeks of alirocumab treatment (*p* < 0.001). The relative increase in plasma triglycerides at 4 h versus t0 was approximately 25%. There was no effect of alirocumab on the post-prandial triglyceride response comparing changes in triglycerides (t4–t0) before and after alirocumab treatment (*p* = 0.446) or comparing the respective areas under the curve (*p* = 0.485, Figure 1 and Appendix A). Total, HDL, and non-HDL cholesterol remained unchanged during the fat-tolerance test.

### 3.4. Effects of Alirocumab on Lipids and Lipoprotein Particles

There was an LDL cholesterol reduction of 69 mg/dL (−44%) in response to treatment with alirocumab. Cholesterol was reduced by 31%, ApoB by 40%, and Lp(a) by 14%. Two patients had documented familial hypercholesterolemia; in one of them, baseline LDL cholesterol was >300 mg/dL. Nine participants had Lp(a) values >50 mg/dL. In this subgroup, Lp(a) was reduced from 78 mg/dL to 68 mg/dL, resulting in a mean absolute reduction of 10 (10) mg/dL (*p* = 0.021). Alirocumab was also associated with a significant increase in HDL cholesterol of +17%. VLDL cholesterol did not change significantly, although a numerical reduction of 34% was observed (Table 2). Alirocumab treatment was associated with an increase in the VLDL triglycerides/VLDL ApoB ratio, which is considered a marker of VLDL size. Nuclear magnetic resonance analysis also suggested a modest increase in VLDL size but no change in the large VLDL particle concentration (Table 3 and Appendix A). There was a significant reduction in the total LDL particle concentration and a significant increase in total HDL particles. The decrease in the total LDL particle concentration was accounted for by a decrease in both large and small particles. The increase in the total HDL particle concentration was accounted for by an increase in buoyant HDL particles (Table 3 and Appendix A).

### 3.5. Effects of Alirocumab on Lipoprotein Lipase Regulators and Circulating PCSK9

Lipoprotein lipase-regulating proteins did not change in response to treatment with alirocumab (Table 2 and Table 3). All patients with self-reported adherence showed a more than three-fold increase in circulating PCSK9 compared with their baseline examination. The plasma PCSK9 level increased from 0.25 to 2.24 µg/mL in the trial completion population, resulting in a mean change of +1.99 µg/mL (SD ± 0.68; *p* < 0.001; *n* = 19). The two non-adherent patients did not show an increase in circulating PCSK9 (<2-fold) and there was no LDL cholesterol reduction. Their PCSK9 levels remained <0.4 µg/mL, whereas, for all other adherent patients, these ranged between 1.1 and 3.8 µg/mL on alirocumab treatment. Among those participants with self-reported adherence, two patients did not show the expected decrease in LDL cholesterol, reflecting low/non-response to alirocumab treatment, but both had a 10- to 12-fold PCSK9 increase (Figure 2 and Appendix A).

## 4. Discussion

The main finding of this pre-specified analysis of the ALIROCKS trial was that triglyceride metabolism was hardly affected by alirocumab treatment in patients with atherosclerosis and hypercholesterolemia. Fat-tolerance testing did not reveal significant effects of alirocumab on post-prandial lipemia. Proteins regulating triglyceride-rich lipoprotein metabolism, such as ApoCII, ApoCIII, GPIHBP-1, ANGPTL-3, and ANGPTL-4, remained unchanged. This supports the hypothesis that PCSK9 does not have strong effects on lipoprotein lipase. We could only detect a modest, non-significant reduction in fasting plasma triglycerides of 10%. This result is consistent with findings of a large, pooled analysis of 10 phase III alirocumab trials with 4983 participants, also reporting a modest triglyceride reduction of approximately 15% [33]. The pooled analysis also suggested that the triglyceride-lowering effect of alirocumab treatment is independent of a metabolic syndrome. However, patients with diabetes mellitus were excluded from this pooled analysis. The patient characteristics of our clinical trial were consistent with a recently published “real-world” non-interventional alirocumab study in clinical routine [34]. Not surprisingly, the effects of PCSK9 antibodies on triglyceride metabolism are more pronounced in patients with insulin-dependent type-2 diabetes and mixed hyperlipidemia, as shown in the ODYSSEY DM-INSULIN study [35]. Another randomized placebo-controlled trial with 12 patients with insulin-dependent type-2 diabetes mellitus also reported that alirocumab treatment reduced both fasting and post-prandial plasma triglycerides [36]. Similar reductions in post-prandial lipemia were reported for the PCSK9 antibody evolocumab in a non-randomized trial with 15 participants suffering from type-2 diabetes [37].

Another objective of the trial was to test the effects of alirocumab treatment on lipoprotein particles. Baseline small dense LDL particle and triglyceride-rich large VLDL particle concentrations showed a strong and positive association, reflecting the features of diabetic dyslipidemia [20]. There was no relevant positive correlation between small and large LDL particles, which is consistent with physiologic considerations and previous clinical studies, supporting the validity of the present nuclear magnetic resonance spectroscopy results [38]. Importantly, treatment with alirocumab was associated with a significant decrease in total, large, and small dense LDL particle concentrations. In contrast, the HDL particle concentration increased. These results are consistent with the findings of a post-hoc analysis of a randomized controlled phase 2 trial with 29 participants [39]. Of relevance, we also confirmed a previous study of our group with 350 participants wherein alirocumab treatment was suggested to be associated with an increase in VLDL size, which was estimated by an increased VLDL triglycerides/VLDL ApoB ratio, and a reduction in VLDL-associated apolipoproteins, suggesting higher clearance of small atherogenic VLDL remnant particles [31]. The present nuclear magnetic resonance spectroscopy data support this hypothesis regarding the increase in VLDL size.

We also observed a significant decrease in Lp(a), which may contribute to cardiovascular risk reduction beyond the beneficial effects of LDL cholesterol lowering, as recently suggested by a pre-specified analysis of the ODYSSEY OUTCOMES study [40]. Whether targeted Lp(a) lowering will reduce the cardiovascular event rate has yet to be demonstrated by an interventional outcome trial that is currently ongoing [41].

A very interesting but preliminary finding of the ALIROCKS study was that the total plasma concentration of PCSK9 may be a possible biomarker for adherence to treatment with PCSK9 antibodies. Circulating PCSK9 after treatment with PCSK9 antibodies comprises unbound PCSK9 and PCSK9 bound to PCSK9 antibodies. The immunoassay used in the ALIROCKS trial does not differentiate between antibody-bound or -unbound PCSK9. In the entire ALIROCKS cohort, total circulating PCSK9 markedly increased in response to alirocumab treatment, most likely due to a strong increase in PCSK9 bound to alirocumab. This finding agrees with the results from a recent study of our group [42]. Of note, two patients only received alirocumab twice and consequently showed neither a reduction in LDL cholesterol nor an increase in circulating PCSK9 at week 10. On the other hand, all patients adherent to alirocumab treatment showed a more than three-fold increase in total circulating PCSK9. Only those two trial participants with <25% LDL cholesterol reduction but with a more than three-fold increase in total circulating PCSK9 may be categorized as low- or non-responders. Therefore, measuring total plasma PCSK9 may be instrumental in categorizing patients with no or low LDL cholesterol reduction as non-adherent patients or non-responders (Table 4). Such adherence testing strategies are increasingly relevant in times of expensive medications and constrained healthcare budgets [43]. However, systematic and larger investigations are required to support or challenge this proposed concept.

The major strength of the ALRIOCKS study is the very precise clinical and metabolic characterization of the study participants, including lipid-tolerance testing. The Lipotest^®^ used in the ALIROCKS study is the best-described, sole diagnostic meal worldwide in commercial form to date [15]. This is also the first clinical trial prospectively investigating the effects of alirocumab on lipoprotein subclass particle concentrations measured with nuclear magnetic resonance spectroscopy in a “real-world” setting.

The limitations of the ALIROCKS trial include the lack of a control group. However, all participants had a medical indication for treatment with PCSK9 antibodies. Hence, it would have been unethical to withhold alirocumab from half of the participants. The relatively small sample size may also be regarded a weak point of the ALIROCKS study. Hence, future, larger studies are encouraged to confirm or challenge the present preliminary results of this exploratory pilot study. Nevertheless, it is unlikely that a clinically relevant impact of short-term lipid-lowering therapy on post-prandial lipemia will be detected in patients without diabetic dyslipidemia considering the minor effect size in the present cohort.

In summary, we could not detect significant effects of alirocumab treatment on triglyceride metabolism or post-prandial lipemia in a patient population without metabolic or glycemic preselection. Remarkably, total circulating PCSK9 may be instrumental in differentiating PCSK9 antibody non-responders from non-adherent patients. Future studies are required to confirm the present preliminary results.

## Figures and Tables

**Figure 1 biomedicines-10-00193-f001:**
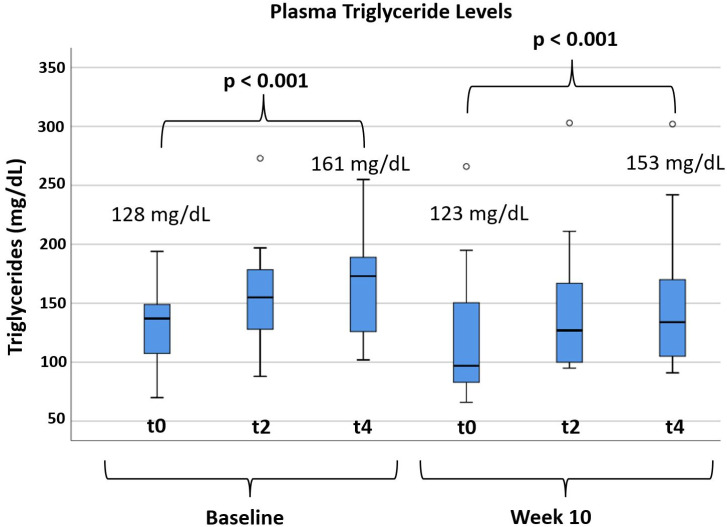
Fat-tolerance test results with Lipotest^®^ meal at baseline and after 10 weeks of treatment with alirocumab. Legend: Triglyceride plasma levels at baseline and after 10 weeks of alirocumab treatment, before Lipotest^®^ meal consumption (t0), at 2 h (t2), and 4 h (t4). Among the trial completion population, 14 patients participated in the per-protocol optional fat-tolerance assessment (*n* = 14). The small circles at the upper part of the figure identify outlier measurements. Outliers are defined as values between 1.5× and 3× interquartile ranges from the end of a box. Paired-samples *t*-test with two-sided *p*-value was performed.

**Figure 2 biomedicines-10-00193-f002:**
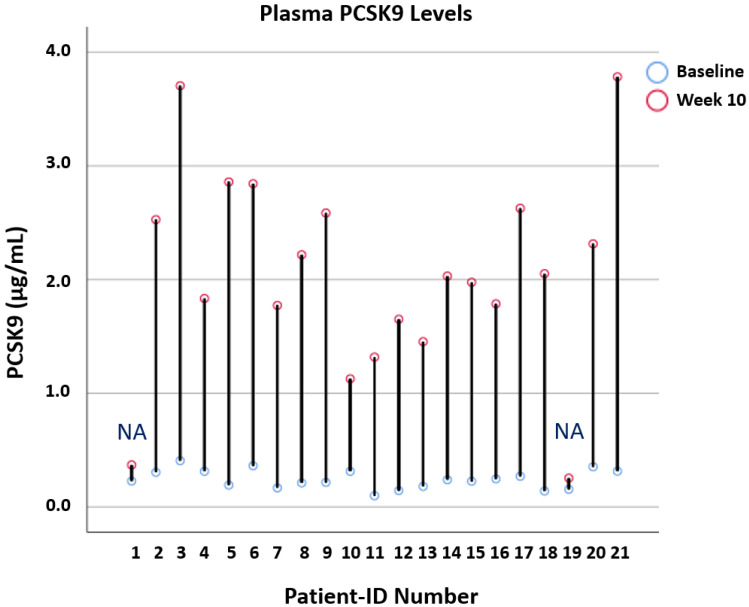
Plasma PCSK9 increase per patient in response to alirocumab treatment. Legend: The figure includes the 19 adherent patients and 2 non-adherent (NA) patients that did not receive alirocumab beyond week 2 (*n* = 21). The used enzyme-linked immunosorbent assay measured total plasma PCSK9 levels, and thus did not differentiate between bound and unbound PCSK9.

**Table 1 biomedicines-10-00193-t001:** Characteristics of trial participants (trial completion).

Characteristic	Alirocumab (N = 19)
Age—year	66 (9)
Female sex—*n* (%)	9 (47.4)
Male sex—*n* (%)	10 (52.6)
Smoker ^a^—*n* (%)	9 (47.4)
Current Smoker ^b^—*n* (%)	4 (21.1)
Concomitant Diseases—*n* (%)	
Cardiovascular Disease	19 (100)
a. Coronary Heart Disease	19 (100)
Coronary Intervention or Surgery	16 (84.2)
Documentation of Coronary Stenosis ^c^	3 (15.8)
b. Peripheral Artery Disease	3 (15.8)
c. Cerebral Artery Disease	6 (31.6)
Chronic Kidney Disease	4 (21.1)
Familial Hypercholesterolemia ^d^	2 (10.5)
Adiposity	4 (21.1)
Type-2 Diabetes Mellitus	4 (21.1)
Type-1 Diabetes Mellitus	0 (0)
Hypertension	15 (78.9)
Number of prior Cardiovascular Events ^e^—*n* (%)	
Three	2 (10.5)
Two	5 (26.3)
One	9 (47.4)
Zero	3 (15.8)
Concomitant Lipid Medication—*n* (%)	
High-Intensity Statins ^f^	3 (15.8)
Statins	5 (26.3)
Ezetimibe	13 (68.4)
Dietary Supplements ^g^	5 (26.3)
Statin Intolerance ^h^	16 (84.2)

**Legend:** Values are numbers (percentages) or means (standard deviations) for categorical and continuous variables, respectively. ^a^ Current or former smoker. ^b^ Documented as current smoker or no stop date documented. ^c^ Confirmed by cardiac computed tomography but without documentation of prior cardiovascular event (e.g., stroke, myocardial infarction, or percutaneous intervention). ^d^ According to medical records. ^e^ Documented as stent, balloon, coronary artery bypass graft, myocardial infarction, or percutaneous intervention, prior strokes/transient ischemic attacks. ^f^ Documented as ≥40 mg of atorvastatin or ≥20 mg of rosuvastatin. ^g^ Exclusively red yeast rice combination products (monacolin K). ^h^ Patients that did not receive high-intensity statins at baseline (includes partial or complete intolerance).

**Table 2 biomedicines-10-00193-t002:** Effects of alirocumab on lipids, lipoprotein particle subfractions, and lipoproteins.

Parameters	Baseline	Week 10	Absolute Change (mg/dL)	±SD	Relative Change (%)	*p*-Value
Cholesterol (mg/dL)	236	162	−74	42	−31	<0.001
Triglycerides (mg/dL)	144	130	−15	69	−10	0.362
Lp(a) (mg/dL)	47	41	−7	8	−14	0.002
ApoAI (mg/dL)	154	164	+10	13	+7	0.002
ApoAII (mg/dL)	34	35	+1	4	+2	0.384
ApoB (mg/dL)	110	66	−44	23	−40	<0.001
ApoCII (mg/dL)	6	5	−1	1	−10	0.069
ApoCIII (mg/dL)	14	14	−1	3	−5	0.411
ApoE (mg/dL)	12	9	−3	3	−21	0.001
VLDL Cholesterol (mg/dL)	35	23	−12	41	−34	0.219
VLDL Triglycerides (mg/dL)	96	91	−5	67	−5	0.772
VLDL ApoB (mg/dL)	15	10	−5	10	−32	0.046
LDL Cholesterol (mg/dL)	158	89	−69	63	−44	<0.001
LDL Triglycerides (mg/dL)	36	24	−11	7	−32	<0.001
LDL ApoB (mg/dL)	95	56	−39	27	−41	<0.001
HDL Cholesterol (mg/dL)	42	49	+7	7	+17	<0.001
HDL Triglycerides (mg/dL)	13	14	+1	2	+9	0.055

**Legend:** Shows lipoprotein analysis results by ultracentrifugation and precipitation method at baseline and after 10 weeks of alirocumab treatment. Trial completion population (*n* = 19). SD: Standard deviation; Mean values; Paired-samples *t*-test with two-sided *p*-value.

**Table 3 biomedicines-10-00193-t003:** Effects of alirocumab on lipoprotein particle numbers, ratios of particle subfractions, and lipoprotein lipase regulators.

Parameters	Baseline	Week 10	Absolute Change	±SD	Relative Change (%)	*p*-Value
Lipoprotein Particle Number ^a^						
L-VLDLp (nmol/L)	6.5	6.9	+0.4	5.6	+6	0.752
LDLp (nmol/L)	1573	930	−643	317	−41	<0.001
L-LDLp (nmol/L)	832	498	−335	454	−40	0.005
S-LDLp (nmol/L)	741	448	−293	301	−40	<0.001
HDLp (nmol/L)	32,391	34,585	+2194	2294	+7	0.001
L-HDLp (nmol/L)	4758	6083	+1325	1382	+28	0.001
S-HDLp (nmol/L)	28,436	28,940	−593	2122	+2	0.252
Lipoprotein Particle Size ^b^						
VLDL size (nm)	49	52	+3	5	+6	0.031
LDL size (nm)	21	21	0	0.5	0	0.159
HDL size (nm)	8.8	9.0	+0.2	0.2	+2.3	<0.001
Ratios of Lipoprotein Particles ^c^						
Triglycerides/ApoB	1.3	2.2	+0.9	0.8	+69	<0.001
VLDL Triglycerides/ApoB	0.9	1.6	+0.7	0.8	+78	0.001
VLDL Triglycerides/VLDL-ApoB	6.9	9.7	+2.8	2.8	+41	<0.001
Cholesterol/ApoB	2.2	2.6	+0.5	0.3	+18	<0.001
Lipoprotein Lipase Regulators ^d^						
ANGPTL-3 (ng/mL)	66.5	66.3	−0.2	4.3	−0.3	0.835
ANGPTL-4 (pg/mL)	138.3	140.6	−2.2	43.1	+1.7	0.825
GPIHBP-1 (pg/mL)	894.3	913.7	+19.4	118.3	+2.2	0.484

**Legend:** Table shows mean values and changes; Trial completion analysis (*n* = 19); Paired *t*-test with two-sided *p*-value. SD: Standard deviation. ^a,b^ Lipoprotein particle number concentration and particle size were assessed by nuclear magnetic resonance spectroscopy. L = large; S = small lipoprotein particles. Total and small VLDL particle concentrations are not provided by the Numares method. ^c^ Ratios of lipoprotein subfractions were calculated from lipoprotein analysis using the combined ultracentrifugation precipitation method. ^d^ Parameters were measured by enzyme-linked immunosorbent assays.

**Table 4 biomedicines-10-00193-t004:** Proposed evaluation for PCSK9 antibody adherence.

	Responder	Non-Responder	Non-Adherent
LDL Cholesterol ↓	Yes	No	No
PCSK9 ↑ (x3)	N.A.	Yes	No

**Legend:** Proposed evaluation strategy for testing low- to non-responder versus non-adherence. Total circulating PCSK9 level may differentiate between non-responder and non-adherent. Suggested cut-off value is a >3-fold PCSK9 level increase from baseline. ↑ = Increase, ↓ = Decrease; N.A.: Not Applicable.

## Data Availability

The data presented in this study are available on specific request from the corresponding author.

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
