# Peer review of "Effects of Alirocumab on Triglyceride Metabolism: A Fat-Tolerance Test and Nuclear Magnetic Resonance Spectroscopy Study"

_biomedicines, 2022, doi:10.3390/biomedicines10010193_

Round 1
Reviewer 1 Report
See the attached file.
In this work, the authors performed a 24-patient ALIROCKS study to investigate whether a 10-week treatment course with 75-150 mg PCSK9 inhibitor Alirocumab could change VLDL-C, aside from the established effects of reducing LDL-C and increasing HDL-C. In the 19 compliant patients, Alirocumab decreased LDL-C and LDL particles in all particle sizes and increased HDL-C, as expected. However, the treatment did not change VLDL-C concentrations, although it may increase large VLDL particles, suggesting that the more atherogenic small VLDL particles. Another important piece of data is that all the lipases or lipase-associated proteins examined (apoCII, apoCIII, GPIHBP-1, ANGPTL-3, ABGPTL-4) were not affected by the treatment. The observations are interesting. The observations further define the therapeutic target of a representative PCSK9 inhibitor. The sample size can be enlarged somewhat but the results will likely remain unchanged. Nevertheless, it will help if other researchers can confirm or challenge these conclusions.

Author Response
Many thanks for your careful review and your positive feedback to our manuscript.
We agree that further studies with larger sample sizes are encouraged to confirm our findings, please also see limitations section, page 10/11, lines 51-54, 1-3:
“The relatively small sample size may also be regarded a weak point of the ALIROCKS study. Hence, future, larger studies are encouraged to confirm or challenge the present preliminary results of this exploratory pilot study. Nevertheless, it is unlikely that a clinically relevant impact of short-term lipid-lowering therapy on post-prandial lipemia will be detected in patients without diabetic dyslipidaemia considering the minor effect size in the present cohort.”
Reviewer 2 Report
- This study investigated the effects of alirocumab on TG metabolism including changes in plasma TG level, TG-rich lipoproteins and lipoprotein lipase regulators. Although the current study has been a pilot trial so far, the reviewer believe that this study would have a potential to provide clinically meaningful information.
- The reviewer’s main concerns are relatively poor description and discussion regarding TG-rich lipoproteins and atherogenic lipoprotein subfractions such as small dense LDL and Lp(a). More detailed explanation for these lipoproteins in combination with TG metabolism should be provided, especially in introduction, to describe background and purpose of this study clearly.
- Page 2, Line 17-18: References regarding small dense LDL should be added. Relationship between small dense LDL and TG-rich LDL should be documented, since it is known that TG degradation in TG-rich LDL leads to production of small dense LDL.
- Page 6, Line 14-15 and page 9, line 18-19: The data for VLDL-Triglyceride/VLDL-apoB is not convincing to justify the increase in particle size of VLDL. Image data for VLDL particle analyzed by Nuclear Magnetic Resonance Spectrometry and comparison of the distribution of VLDL particle size would significantly strengthen the results.
- There is no explanation for the function of ANGPTL-3, -4 and GPIHBP-1 to describe their roles in TG-rich lipoprotein metabolism. Brief explanation should be added for readers who are not familiar with vascular TG metabolism.
- Page 3, Line 17-20: Appropriate references regarding the method for fractionation of lipoproteins with combination of precipitation should be added.
- The reviewer concerns limited samples in this study, even if a pilot trial at present. Please explain whether the current sample size is efficient as a pilot trial.
Author Response
- This study investigated the effects of alirocumab on TG metabolism including changes in plasma TG level, TG-rich lipoproteins and lipoprotein lipase regulators. Although the current study has been a pilot trial so far, the reviewer believe that this study would have a potential to provide clinically meaningful information.
Response
Many thanks for your careful evaluation of our manuscript and the positive feedback.
- The reviewer’s main concerns are relatively poor description and discussion regarding TG-rich lipoproteins and atherogenic lipoprotein subfractions such as small dense LDL and Lp(a). More detailed explanation for these lipoproteins in combination with TG metabolism should be provided, especially in introduction, to describe background and purpose of this study clearly.
Response
We have now revised the introduction and discussion section with a special focus on triglyceride-rich lipoproteins and other atherogenic lipoprotein subfractions such as Lp(a) and small-dense LDL particles.
- Please see introduction section page 2, the following sentences were added to better describe the background und purpose of the study:
Page 2, lines 9-16:
“Elevations of circulating triglyceride-rich lipoproteins are the result of their overproduction in the liver and / or their reduced clearance from the circulation. As apolipoprotein B (ApoB) containing lipoproteins, they carry considerable amounts of cholesterol and can enter the subintimal space, especially as remnants after hydrolysis of triglycerides by lipoprotein lipase. The subintimal deposition of cholesterol then leads to atherosclerotic plaques. Fibrates, which reduce triglyceride-rich lipoproteins via activation of peroxisome proliferator activated receptor α have been effective in reducing cardiovascular risk in patients with high triglycerides.”
Page 2, lines 20-30:
“These key regulators include apolipoprotein C-III (ApoCIII), a potent inhibitor of lipoprotein lipase, and its antagonist apolipoprotein C-II (ApoCII). Angiopoietin-like proteins 3 and 4 (ANGPTL-3/4) inhibit lipoprotein lipase forming complexes with angiopoietin-like protein 8 (ANGPTL-8), while glycosylphosphatidylinositol-anchored high density lipoprotein-binding protein-1 (GPIHBP-1) facilitates the transfer of lipoprotein lipase to the luminal side of the endothelium thereby increasing its catalytic activity to hydrolyze plasma triglycerides.
In addition to triglyceride-rich lipoproteins, high concentrations of lipoprotein a (Lp(a)) make up a considerable proportion of residual cardiovascular risk. This evidence is based on very solid genetic evidence from Mendelian randomization studies as the plasma concentration of this small lipoprotein is highly heritable.”
Page 2, lines 32-36:
“In this respect, a high concentration of small-dense LDL particles abundant in diabetic dyslipidaemia is considered atherogenic as they can easily penetrate the endothelium to be deposited in the subintimal space. On the other hand, high concentrations especially of small high density lipoprotein (HDL) particles are considered protective as they are considered effective vehicles for reverse cholesterol transport.”
- Please also see discussion section:
Page 10, lines 4-9:
“Baseline small-dense LDL particle and triglyceride-rich large VLDL particle concentration showed a strong and positive association reflecting the features of diabetic dyslipidaemia. There was no relevant positive correlation between small and large LDL particles, which is consistent with physiologic considerations and previous clinical studies sup-porting the validity of the present nuclear magnetic resonance spectroscopy results.”
Page 10, lines 17-23:
“The present nuclear magnetic resonance spectroscopy data support this hypothesis regarding the increase of VLDL size.”
“We also observed a significant decrease of Lp(a), which may contribute to cardiovascular risk reduction beyond the beneficial effects of LDL-cholesterol lowering as recently suggested by a pre-specified analysis of the ODYSSEY OUTCOMES study. Whether targeted Lp(a) lowering will reduce cardiovascular event rate has yet to be demonstrated by an interventional outcome trial that is currently ongoing.”
- Page 2, Line 17-18: References regarding small dense LDL should be added.
Response
References regarding small-dense LDL were included, please see page 2, lines 32-34:
“In this respect, a high concentration of small-dense LDL particles abundant in diabetic dyslipidaemia is considered atherogenic as they can easily penetrate the endothelium to be deposited in the subintimal space.”1,2,3
- Meeusen JW. Is Small Dense LDL a Highly Atherogenic Lipid or a Biomarker of Pro-Atherogenic Phenotype? Clin Chem. Jul 6 2021;67(7):927-928.
- Hoogeveen RC, Gaubatz JW, Sun W, et al. Small dense low-density lipoprotein-cholesterol concentrations predict risk for coronary heart disease: the Atherosclerosis Risk In Communities (ARIC) study. Arterioscler Thromb Vasc Biol. May 2014;34(5):1069-1077.
- Krauss RM. Lipids and lipoproteins in patients with type 2 diabetes. Diabetes Care. Jun 2004;27(6):1496-1504.
- Relationship between small dense LDL and TG-rich LDL should be documented, since it is known that TG degradation in TG-rich LDL leads to production of small dense LDL.
Response
We have conducted additional magnetic resonance spectroscopy analysis to support this relationship and make this clear to the reader.
Please see results section, page 5, lines 22-24:
“There was a significant positive association of baseline small-dense LDL particles and triglyceride-rich large VLDL particles (r = 0.883, p < 0.001). Small and large LDL particles were not significantly correlated (r = 0.021, p = 0.932).”
Please also see methods section, page 4, lines 9-11:
“Total and small VLDL particle concentrations are not provided by the Numares method (AXINON® lipoFIT® nuclear magnetic resonance assay, Supplementary Materials).”
Please also see discussion section, page 10, lines 4-9:
“Baseline small-dense LDL particle and triglyceride-rich large VLDL particle concentration showed a strong and positive association reflecting the features of diabetic dyslipidaemia. There was no relevant positive correlation between small and large LDL particles, which is consistent with physiologic considerations and previous clinical studies supporting the validity of the present nuclear magnetic resonance spectroscopy results.”
- Page 6, Line 14-15 and page 9, line 18-19: The data for VLDL-Triglyceride/VLDL-apoB is not convincing to justify the increase in particle size of VLDL. Image data for VLDL particle analyzed by Nuclear Magnetic Resonance Spectrometry and comparison of the distribution of VLDL particle size would significantly strengthen the results.
Response
We have now added additional nuclear magnetic resonance analysis which seems to support our preliminary findings. At the same time, we underlined the exploratory character of our pilot trial and encouraged the readership to challenge our study results.
Please see results section, page 6, lines 22-26:
“Alirocumab treatment was associated with an increase of the VLDL-triglycerides/VLDL-ApoB ratio which is considered a marker of VLDL size. Nuclear magnetic resonance analysis also suggested a modest increase of VLDL size but no change of the large VLDL particle concentration (Table 3 and Figure S2).”
Beside Table 3. (e.g., VLDL-size), please also see supplementary materials, Figure S2, page 6: “Distribution of VLDL particle size and changes by alirocumab treatment.”
Please also see discussion section, page 10, lines 12-18:
“Of relevance, we also confirmed a previous study of our group with 350 participants that alirocumab treatment may be associated with an increase of VLDL size, which was esti-mated by an increased VLDL-triglycerides/VLDL-ApoB ratio, and a reduction of VLDL-associated apolipoproteins suggesting a higher clearance of small atherogenic VLDL remnant particles. The present nuclear magnetic resonance spectroscopy data support this hypothesis regarding the increase of VLDL size.”
- There is no explanation for the function of ANGPTL-3, -4 and GPIHBP-1 to describe their roles in TG-rich lipoprotein metabolism. Brief explanation should be added for readers who are not familiar with vascular TG metabolism.
Response
We have now included a brief description of these lipase regulators in the introduction section as suggested.
Please see page 2, lines 20-26:
“These key regulators include apolipoprotein C-III (ApoCIII), a potent inhibitor of lipoprotein lipase, and its antagonist apolipoprotein C-II (ApoCII). Angiopoietin-like proteins 3 and 4 (ANGPTL-3/4) inhibit lipoprotein lipase forming complexes with angiopoietin-like protein 8 (ANGPTL-8), while glycosylphosphatidylinositol-anchored high density lipoprotein-binding protein-1 (GPIHBP-1) facilitates the transfer of lipoprotein lipase to the luminal side of the endothelium thereby increasing its catalytic activity to hydrolyze plasma triglycerides.”
- Page 3, Line 17-20: Appropriate references regarding the method for fractionation of lipoproteins with combination of precipitation should be added.
Response
We have now renamed this analysis method to: “Lipoprotein Analysis by Ultracentrifugation and Precipitation” due to minor deviations to other published quantification methods (ß-Quantification), please see methods section, page 3, lines 32-34. The method was renamed accordingly throughout the manuscript as well as in all table and figure legends.
Additionally, we have now included appropriate references from older originator publications as well as a newer reference, please see page 3, line 34:
- Wanner C, Horl WH, Luley CH, Wieland H. Effects of HMG-CoA reductase inhibitors in hypercholesterolemic patients on hemodialysis. Kidney Int. Apr 1991;39(4):754-760.
- Bachorik PS, Ross JW. National Cholesterol Education Program recommendations for measurement of low-density lipoprotein cholesterol: executive summary. The National Cholesterol Education Program Working Group on Lipoprotein Measurement. Clin Chem. Oct 1995;41(10):1414-1420.
- Silbernagel G, Scharnagl H, Kleber ME, et al. LDL triglycerides, hepatic lipase activity, and coronary artery disease: An epidemiologic and Mendelian randomization study. Atherosclerosis. Mar 2019;282:37-44.
- The reviewer concerns limited samples in this study, even if a pilot trial at present. Please explain whether the current sample size is efficient as a pilot trial.
Response
We fully acknowledge the exploratory design of our pilot study including the small sample size. This has been made clear to the reader throughout the manuscript and most important at the following sections:
Please see methods section, statistical analysis, page 4, line 13:
“The statistical design of this pilot trial was explorative and descriptive.”
Please also see limitations section, page 10, lines 51-52:
“The relatively small sample size may also be regarded a weak point of the ALIROCKS study.”
Also taking the comments of Reviewer 1 into account, we have also included the following sentences in the limitations section, please see page 10/11, lines 53-54/1-3:
“Hence, future, larger studies are encouraged to confirm or challenge the present preliminary results of this exploratory pilot study. Nevertheless, it is unlikely that a clinically relevant impact of short-term lipid-lowering therapy on post-prandial lipemia will be detected in patients without diabetic dyslipidaemia considering the minor effect size in the present cohort.”
